# Testing the effectiveness of a weight loss intervention to enhance self-regulation in adults who are obese: protocol for a randomised controlled trial

Kerstin Frie 🄳 , Jamie Hartmann-Boyce 🄳 , Susan A Jebb 🄳 , Paul Aveyard 🄳

Nuffield Department of Primary Care Health Sciences, University of Oxford, Oxford, UK

**Correspondence to**
Kerstin Frie;
kerstin.frie@phc.ox.ac.uk

## ABSTRACT

**Introduction** Previous trials finding an effect of self-monitoring on weight loss have considered the effect to be mediated by self-regulatory processes. However, a qualitative think-aloud study asking people to record thoughts and feelings during weighing showed that self-regulation occurs only rarely without further instruction. The aim of this trial is to test a novel intervention guiding people through the self-regulatory processes to see whether it facilitates weight loss.

**Methods and analyses** A parallel group, randomised controlled trial will be conducted to test the concept that a self-regulation intervention for weight loss increases weight loss compared with daily self-weighing without further support. One hundred participants with a body mass index ≥30 kg/m$^2$ will be randomised to either the control or intervention group. The control group will be asked to weigh themselves daily for 8 weeks, the intervention group will be encouraged to follow the self-regulation intervention. They will be prompted to weigh daily, track their weight using an app, plan daily actions for weight loss and reflect on their action plans on a weekly basis. This self-regulation cycle will allow them to experiment with different weight loss strategies and identify effective and sustainable actions. Primary and process outcomes will be measured at baseline and 8 weeks' follow-up. Linear regression analysis of the primary outcome, weight change, will assess the early effectiveness of the intervention. The process outcomes liking, perceived effectiveness, as well as usage and barriers with regard to the self-regulation intervention, will be assessed through qualitative analysis of follow-up interviews and quantitative analysis of adherence rates and responses to a final questionnaire.

**Ethics and dissemination** This trial was reviewed and approved by the NHS National Research Ethics Committee and the Health Research Authority (reference number: 18/SC/0482). The findings of the trial will be published in peer reviewed journals and presented at conferences.

**Trial registration number** ISRCTN14148239, prerecruitment.

**Protocol version** Version 1.1, 7 December 2018.

## Strengths and limitations of this study

► This trial will provide preliminary evidence for whether a remotely delivered and iterative self-regulation intervention can enhance weight loss.
► The intervention addresses specific barriers to self-regulation, identified in a previous qualitative study of experiences of self-weighing.
► Process evaluation measures will assess participants liking, usage and perceived effectiveness of the intervention to enable future improvements.
► Blinding might be compromised as participants might be aware that they were assigned to the control or treatment group.
► The study is designed as a 'proof-of-concept' trial and longer-term studies will need to evaluate the effectiveness of the intervention as a weight management tool.

## INTRODUCTION

Self-monitoring of weight is often employed in multicomponent weight loss interventions, as evidence suggests it aids weight loss.[1–5] The weight loss effect has often been ascribed to a self-regulation mechanism, based on the hypothesis that self-monitoring triggers self-regulation.[1 4–6] In this context, self-regulation occurs in iterative cycles, starting by (1) contextualising the weight with previous measurements and goals, thus providing (2) an opportunity to reflect on previous behaviour and reinforce successful actions, enabling (3) the planning of actions to reach the goal, followed by (4) the performance of planned actions.[6–8] This cycle of processes allows for experimentation with different weight loss techniques, helping the user to build a personal portfolio of effective and sustainable strategies.[9 10]

A study employing self-weighing as a stand-alone intervention did not find a significant

weight loss effect, raising the question whether self-regulation is performed naturally after weighing or whether additional weight loss treatment components are necessary.[11] We addressed this question in a think-aloud study, where twenty-four participants were asked to record their thoughts and feelings during daily weighing for 8 weeks, without being prompted to self-regulate.[12] On 90% of occasions, participants contextualised their weight measurement, and on 58% participants reflected on previous behaviours. Only on 20% of occasions did participants plan actions and specific action planning, defining a concrete action and time plan, was rare (6%). The frequency of specific action planning was, however, significantly predictive of weight loss. Hence, the study provided support to the notion that completing the last step of the self-regulation process can elicit weight loss. However, the think-aloud study also showed that self-regulation does not occur autonomously, and that people need support in developing self-regulation skills, especially action planning. Self-regulation, once learnt, has the potential to be a weight loss strategy that is performed autonomously and sustainably. It provides an opportunity for remote weight loss interventions that do not require resource-intensive support from healthcare providers.

Several weight loss interventions including self-regulation components have been developed over the years. Some educate their users about self-regulation theory without providing active support for the key component—specific action planning.[1 4 13 14] Others support users in action planning, but fail to imitate the iterative nature of self-regulation, as they do not allow the usage of self-monitoring feedback for adaptations to the action plans,[15–18] thus disabling self-regulation. Some interventions guide participants through iterative reformulation of action plans following self-monitoring feedback, but are delivered through face-to-face sessions,[19 20] which are resource-intensive and not easily rolled out at large scale. Other interventions incorporating action planning dictate which actions participants are supposed to follow,[15 21 22] which might reduce goal ownership, a significant predictor of goal engagement and attrition.[23 24] Notwithstanding these critiques, seven of the eleven studies cited here found significant weight loss, suggesting that components targeting the self-regulation cycle can enhance weight loss, encouraging further work in this area. Our critiques highlight the need for an intervention that guides participants through the whole and iterative self-regulation process in an autonomous, low-cost and scalable manner. Furthermore, since many interventions add self-regulation elements to a broader spectrum of weight loss treatment components,[14 17 25–27] a study testing self-regulation as a standalone intervention is needed to investigate whether iterative self-regulation is sufficient to achieve weight loss.

With this study we aim to test the proof of concept of an intervention aiming to address this gap in the literature. The PREVAIL intervention (*P*eople *RE*gulating themsel*V*es to *A*chieve we*I*ght *L*oss) is a weight loss programme guiding people through the iterative self-regulation process. It encourages users to experiment with different weight loss approaches, and use the self-regulation mechanism to find their ideal set of tools.

## Objectives

The primary objective of this trial is to test the concept whether an intervention which trains individuals in self-regulatory processes, aids early weight loss in comparison to unsupported daily weighing. Other objectives pertain to the evaluation of usage and effectiveness of the self-regulation intervention components, as well as the qualitative analysis of participant experiences of the intervention.

## METHODS

### Study design and setting

An individually randomised, two arm, parallel group design will be employed, assessing superiority of the self-regulation intervention over daily self-weighing alone. Participation will last 8 weeks. This length was deemed sufficient to assess early effectiveness of the intervention, as previous studies have been able to detect weight loss effects after 2 months.[28] If the results are promising, the data will provide good evidence to justify conducting a longer-term randomised controlled trial. Participants will attend two study visits, one at baseline and one after the end of the eighth week of the intervention. The primary outcome will be weight change. The study will take place in Oxfordshire, UK, and run between April and October 2019.

### Recruitment

Two to four general practitioner (GP) practices around Oxford, UK, will function as participant identification centres and search their health records to identify suitable patients for the trial (age ≥18 years, body mass index (BMI) ≥30 kg/m$^2$). The GP will screen the search list and exclude patients who would be inappropriate to invite, including terminally ill or violent patients. Suitable patients will be sent an invitation letter from their GP. They will be encouraged to contact the research team if they are interested in taking part. GPs may also identify suitable patients during routine consultations. We will ask practices not to refer participants to commercial weight loss programmes, other obesity clinics or bariatric surgery, while they are enrolled as participants in this trial.

### Eligibility criteria

#### Inclusion criteria

► Participant is willing and able to give informed consent.
► Aged 18 years or above.
► BMI≥30 kg/m$^2$.
► Owns an Apple or Android smartphone.

#### Exclusion criteria

The participant may not enter the study if any of the following apply:

- ► Unable to understand English.
- ► Unable to follow all intervention procedures for a period of more than four consecutive days.
- ► Currently self-monitoring body weight more than once a week.
- ► Currently or within 3 months of study entry attended a weight management programme or currently participating in another weight loss study.
- ► Lost more than 5% of current body weight in the last 6 months.
- ► Prior bariatric surgery or scheduled for bariatric surgery.
- ► Pregnant or planning to become pregnant during the course of the study.
- ► Have an electronic medical implant, such as a pacemaker.
- ► Have ever had or been diagnosed with an eating disorder.
- ► People that the GP judges not able to meet the demands of either treatment programme or measurement schedule. This may include severe medical problems not listed above.

## Participant flow
### Screening
People who are interested in taking part will contact the research team. The research team will then discuss study participation by telephone or email and undertake screening. If the person appears eligible and would like to attend a baseline visit, the research team will offer an appointment at a local venue. The participants will be emailed a participant information sheet (PIS).

### Baseline
When the participants attend the baseline appointment, a member of the research team will seek informed consent and check eligibility for inclusion in the study by measuring height and weight for BMI calculation. The participants will be asked to complete an online questionnaire, capturing demographics (ie, age, gender, ethnicity) and previous experiences with self-weighing. Participants will be randomised and receive instructions for the assigned intervention. The researcher will provide participants with a body scale for the duration of the trial. A follow-up appointment will be scheduled for after the completion of the intervention period.

### Follow-up
The aim of the follow-up appointment is to assess the outcomes of the trial. The research team will email participants in advance to remind them of the meeting. In the intervention group, this email will also contain a final questionnaire, asking participants about the usefulness of each of the intervention components and an overall rating of the intervention.

The appointment will be scheduled at a local venue and conducted by a member of the research team.

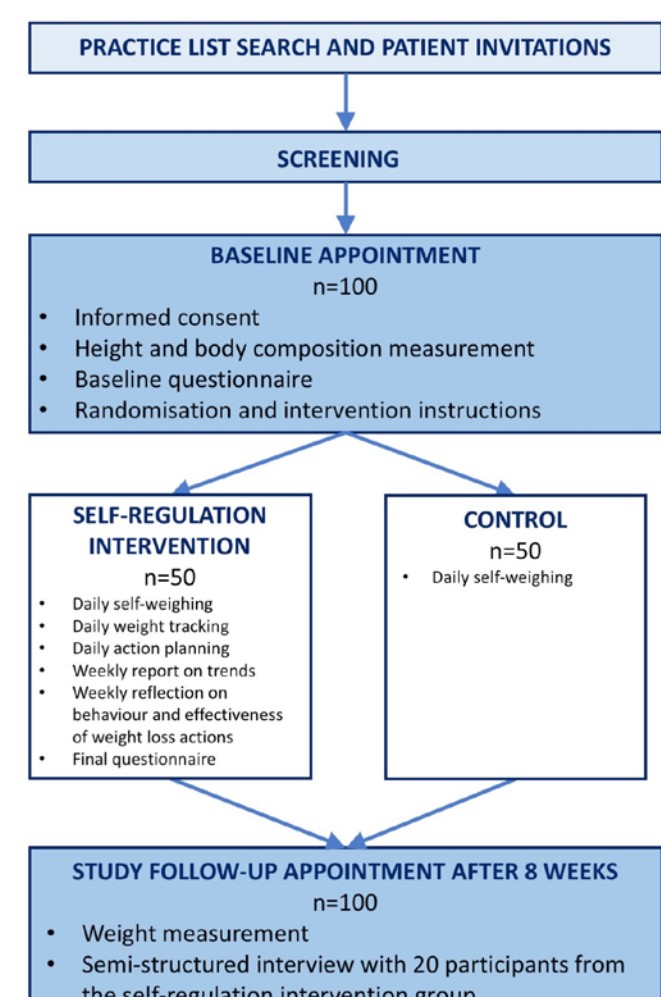

**Figure 1** Study flow chart.

Participants will be asked to return the body scales. The researcher will measure participants' weight.

Twenty participants in the intervention group will be invited to participate in a semi-structured interview at the end of the follow-up meeting. The interviewees will be sampled by the lead researcher with the aim to reflect different levels of adherence, weight change and responses to the final questionnaire. Participants will be asked about their experiences with self-weighing, their liking of the intervention components and perceived barriers to engaging with the intervention. Interviews will be recorded, transcribed and analysed. A study flow chart is displayed in figure 1.

## Sample size
We intend to recruit 100 participants, which is sufficient to detect a 1.5 kg difference between conditions, at 90% power and 5% type I error rate, while allowing for a 20% drop-out rate. Variance of weight change was based on results of a similar trial,[28] which reported a SD of 2.13 kg at 2 months' follow-up.

## Randomisation and blinding

All eligible participants will be randomised with an allocation ratio of 1:1 to the intervention or control group. A randomisation sequence, stratified by GP and using block randomisation with randomly varying block sizes of 2 and 4 will be generated using a computer algorithm. Allocations will be concealed in numbered, sealed, opaque envelopes by an independent researcher in the department and handed to the researcher who will conduct the baseline visits.

Due to the nature of this trial, it will be difficult to blind participants to the treatment allocation. We will aim to make it as opaque as possible by presenting daily weighing as an intervention to control group participants, as done in previous trials.[29] The researchers conducting baseline and follow-up will perform data analysis, and will not be blinded to treatment allocation. The primary outcome, weight change, will be measured objectively. Adherence to self-regulation steps will be measured objectively through the frequency of weight logs and completed questionnaires in control and intervention group. The evaluations of treatment components in the final questionnaire will be measured without researchers' input and analysed quantitatively. Blinding of the researcher who conducts and analyses the semi-structured interviews with intervention group participants will not be possible.

## Intervention and control
### Intervention

Participants will be asked to weigh themselves every morning after waking using standard body scales (Etekcity, California, USA) provided to them. In addition, they will be prompted to complete tasks which speak to the different steps of the self-regulation process, including (1) contextualising the weight measurement, (2) reflecting on behaviour and (3) planning weight loss actions. The individual tasks and their development are described in more detail below. Input from members of the public was sought at several stages of the intervention development.

1. Contextualising: In the think-aloud study participants struggled memorising daily measurements and keeping an overview of their weight loss progress, which impeded their ability to use weight measurements as constructive feedback. Participants stated they would have benefited from a weight tracking tool. Therefore, to support participants in contextualising their weight measurements, we will encourage them to use the app 'Weight Loss Tracker, BMI' by aktiWir GmbH. Research shows that digital tracking devices can significantly increase adherence to self-monitoring,[30 31] perhaps because the visualisation of progress and feedback on weight loss success provides motivation and keeps users on track with their goals.[32]

2. Reflecting: The think-aloud study showed that participants struggled to interpret day-to-day weight changes due to daily fluctuations that were not caused by fat loss or gain.[12] We therefore decided to encourage reflection on a weekly rather than daily basis. Participants will receive weekly emails with feedback on their weekly weight trends, asking them to complete an online questionnaire (Qualtrics, USA). The questionnaire will prompt reflection on the relationship between behaviours performed throughout the week and weight change observed. Participants will be asked to use this insight to evaluate the use of the weight loss actions they had performed throughout the week.

3. Action Planning: We aimed to strike a balance between ensuring that participants choose appropriate actions and allowing them to choose actions themselves, as lack of goal ownership predicts attrition.[23] Participants will therefore choose one weight loss action per day from a list of 53 actions (see online supplementary appendix 1). To create the actions list, we first identified weight loss actions from effective weight loss interventions in the literature. These were reviewed, adapted and complemented during iterative brainstorming sessions with an interdisciplinary expert team, comprised of dietitians, GPs and psychologists. The rationale for daily action plans was twofold: (1) allowing participants to adapt their actions flexibly to their day and (2) giving participants exposure to a wider range of strategies. Based on action planning and implementation intention research, which shows that the specificity of action plans increases likelihood of implementation,[33–35] we will ask participants to specify where, when and how they will perform their chosen action, which cues they will use, which barriers they might experience and how they will deal with them.

As participants only reflect on their behaviour on a weekly basis, we grouped the actions conceptually into seven categories, five of which cover diet-related actions and two of which cover physical activity-related actions. Participants will be asked to choose one category at the beginning of the week, and choose daily actions from this category for the rest of the week. Weekly behaviour reflection will therefore focus on the effectiveness of a category of actions. Taken together, the reflection and action planning process shall enable participants to experiment with different weight loss approaches and decide on their effectiveness and usefulness based on trends in weight data. At the beginning of weeks two through eight of the intervention, participants will be prompted to commit to some of the actions of previous weeks, as the performance of several weight loss actions increases chances for weight loss. Action planning support will be provided through daily questionnaires (Qualtrics, USA), which will be sent to participants every morning via email.

At baseline, participants will receive detailed instructions about the intervention with the aid of a manual which they may take home afterwards. The manual will explain the rationale of self-regulation and experimentation to participants and give tips on how to best weigh on a daily basis. It will also provide information on the different weight loss actions participants can try throughout the study. Participants will also be given an

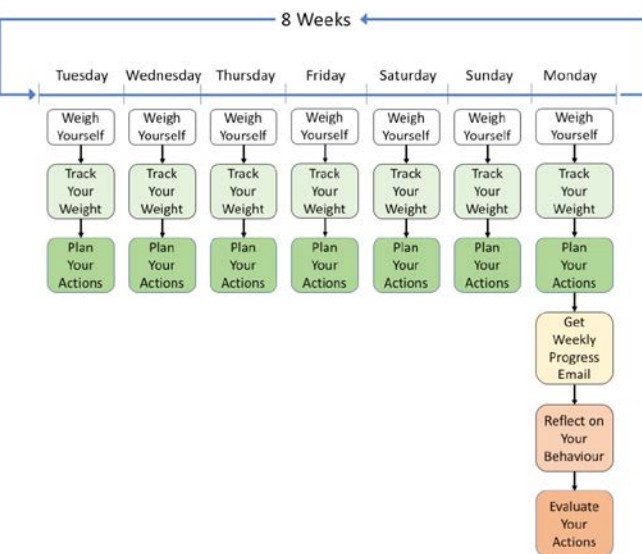

**Figure 2** Intervention procedure as depicted in study manual.

action diary, in which they will be asked to record their performed weight loss actions.

We will call participants at the end of the first and fourth week to ask about and solve any technical problems that may have arisen (eg, with the functioning of the scales or the receiving of questionnaires). A figure from the study manual depicting the intervention procedure is displayed in figure 2. The Template for Intervention Description and Replication (TIDieR) checklist[36] for the PREVAIL intervention and the comparator is reported in table 1.

### Control
Participants in the control group will be instructed to self-weigh daily and see what daily weighing motivates them to do. They will not receive any further instructions. By using this comparator group, we want to test whether the self-regulation process can enhance self-weighing to be an effective weight loss tool. Participants will receive smart scales (BodyTrace, New York) which are equipped with a SIM card and automatically transfer measurements to a secure server via the 3G/4G network. This will allow us to assess adherence to daily weighing in the control group.

### Patients and public involvement
Members of the public were involved in the design of the study at several stages. After creating the invitation letter, PIS, informed consent form, semi-structured interview guide and the baseline, daily, weekly and follow-up questionnaires for the PREVAIL intervention, we asked members of the public for feedback. Our department has a panel of >100 members of the public with an interest in weight management. Based on phone calls with members of this panel we were able to make the materials clearer and more concise. The manual, explaining the intervention in detail, was further discussed with members of the panel in a focus group session. As a result of this focus

group session, the manual was professionally edited to be shorter. We also added figures and graphs to present the procedures of the study more visually. The panel helped to revise the explanations of the different action plans.

A test run of 4 weeks was conducted with five members of the department, who are not otherwise involved in this study. They provided feedback on the running of the intervention which, among other outcomes, led to the creation of a reminder email which will be sent out before the start of the intervention.

No members of the public will be involved in conducting or analysing the study. However, we will gather input on the most suitable and effective ways to disseminate our research findings to the public.

### Outcomes
#### Primary outcome
► Change in body weight between baseline and follow-up by condition.

#### Secondary outcomes (process evaluation)
► Adherence to self-regulation steps, assessed through weight records and daily action planning/weekly evaluation questionnaires.
► Test moderators of effectiveness: adherence measures, highest educational qualification, liking of weighing at baseline, overall rating of intervention in final questionnaire.
► Perceived effectiveness of intervention and liking of intervention features based on final questionnaire and follow-up interviews.
► Barriers and unmet needs for successful weight loss, assessed in follow-up interviews.

### Measurements
A schedule of measurements can be found in table 2.

#### Physical measurements
Participants' height will be measured at baseline to the nearest 0.1 cm using a stadiometer. Weight will be measured both at baseline and follow-up using a digital scale (SC-240 MA, Tanita Japan). Weight will be recorded to the nearest 0.1 kg.

#### Process evaluation measures
##### Adherence measures
For the control group, adherence to daily weighing will be measured by calculating the proportion of days for which we have a recorded weight measurement on the BodyTrace server.

For the intervention group, adherence to daily weighing will be assessed by calculating the proportion of days for which a weight was recorded in the weight tracking app or in the daily action planning questionnaire. Adherence to weight-tracking will be calculated as the proportion of days for which a weight measurement was recorded in the app. Adherence to action planning and reflection will be measured by calculating the proportion of days on which the respective questionnaires were completed. An overall

**Table 1** TIDieR checklist describing the intervention and control condition

| | Intervention: PREVAIL | Control: daily self-weighing without behavioural support |
|---|---|---|
| Brief name | PREVAIL (*People Regulating Themselves to Achieve Weight Loss*) | Self-weighing only |
| Why | *Self-weighing:* Monitoring weight on a daily basis will enable participants to take note of their weight loss progress. *Weight tracking:* Our preceding study[12] found that people can lose track of their weight loss progress when weighing every day because they struggle to remember measurements. We therefore ask participants to track their weight. *Action planning:* In our preceding study, participants rarely made action plans to help them progress with their weight loss.[12] If they did make action plans, they were rarely specific. Specificity of action plans is a significant predictor of the likelihood of implementation.[33–35] We therefore guide participants through a specific action planning process. *Report email:* Participants in our preceding study struggled to see trends in their weight data.[12] Unfortunately, the app we are using for weight tracking does not provide users with trend information. We will therefore send out weekly emails, containing information on the trend of the weight measurements of the last week. *Reflection and action evaluation:* Our preceding study revealed that daily fluctuations vary over time within people, making it difficult to interpret daily weight changes. We therefore want to encourage participants to reflect on their weight changes and the effectiveness of their weight loss actions on a weekly basis. Using the weekly weight trend information, participants will be able to evaluate whether they found the group of actions they performed effective and worth repeating. | A previous trial has found that self-monitoring of weight without further guidance is not effective for weight loss.[11] |
| What | *Self-weighing:* Participants will be instructed to weigh themselves daily using simple digital body scales (Etekcity Corporation, California). They will be asked to weigh themselves in a similar state every day, ideally first thing in the morning and without clothes. *Weight tracking:* Participants will be asked to download the free app "Weight Loss Tracker, BMI" by aktiWir GmbH on their smartphone and use it on a daily basis to record their weight measurements. They will be asked to submit a backup of their data to the research team every week. *Action planning:* Participants will receive a daily questionnaire helping them to plan a weight loss action. The questionnaire will start by asking participants to enter their morning weight. At the beginning of first week, they will then be asked to choose a category of actions, and they will be able to choose one of the actions within this category per day for the rest of the week. There are seven categories in total, five covering diet-related actions and two covering physical activity-related actions. Participants will be asked to specify for each action how, when and where they are going to perform it, and which cues they are going to use. They will also be prompted to think about how to overcome potential barriers. At the start of weeks 2 through 8, participants will be encouraged to try out a new category of actions. We will additionally ask participants on a weekly basis to commit to continuing some of the actions they tried out in previous weeks. In order to help participants to maintain an overview of the actions they performed so far, we will provide them with a non-digital action plan diary. *Report email:* Once a week, participants will receive an email from the research team, informing them about their trend weight change for the last week. This trend weight change consists of the slope of a trend line fitted across all measurements of the week, multiplied by the number of days covered by the measurements. The weight measurements used to create these reports will be taken from the daily action planning questionnaires. *Reflection and action evaluation:* In the weekly report email, participants will receive a link to the reflection and action evaluation questionnaire. This questionnaire prompts participants to think about why their weight has changed as it has. They are further asked to evaluate the group of actions they performed across the week, including whether they found them useful and whether they would repeat them. On the basis of this evaluation, participants will be able to decide which actions they want to continue doing in the next weeks, and which ones to drop. Using this method of self-experimentation, participants will be able to try and test different weight-loss strategies and identify the ones that are effective and sustainable for them. | Self-weigh every morning in a similar state, using smart scales (BodyTrace, New York). |
| Who | The chief investigator (KF) will deliver the intervention at the baseline session. She will also organise the mailing of all questionnaires and weekly report emails. She will be the primary contact for all participants. KF is a psychologist by background and has received Good Clinical Practice (GCP) training. | The chief investigator (KF) will instruct participants in the control group to weigh themselves every day. |
| How | Participants will measure their weight on provided body scales. They will use a free weight tracking app called 'Weight Loss Tracker, BMI' by aktiWir GmbH on their smartphone to track their weight loss progress. All questionnaires will be sent automatically to participants in the early morning by the survey platform Qualtrics (USA). | Participants measure their weight on provided body scales. |
| Where | At home. | At home. |
| Tailored | Participants are able to tailor the intervention to themselves by choosing action plans relevant to them. | N/A |
| How well | We will request participants to send us a back-up of their data in the weight tracking app. This data will allow us to assess adherence to weight tracking. Completion of the daily action planning and weekly reflection questionnaires will allow us to assess adherence to action planning and reflection. Adherence to daily weighing will be assessed by combining weight records from the weight-tracking app and action planning questionnaires. | The provided body scales contain a SIM card which automatically transfers the weight data to a secure research server. We will therefore be able to assess adherence. |

**Table 2** Schedule of measurements

|  | Screening | Baseline visit | Intervention period | Completion email | Follow-up visit (after 8 weeks) |
|---|---|---|---|---|---|
| Length | 10 min | Up to 1 hour | 10 min per day | 5 min | up to 45 min |
| Who conducts | Research team | Research team | Participants | Research team | Research team |
| Eligibility assessment | ☑ | ☑ (BMI) |  |  |  |
| Enrolment |  | ☑ |  |  |  |
| Baseline questionnaire |  | ☑ |  |  |  |
| Weight |  | ☑ | ☑ |  | ☑ |
| Height |  | ☑ |  |  |  |
| Allocation |  | ☑ |  |  |  |
| Weight tracking (only intervention) |  |  | ☑ |  |  |
| Daily questionnaire (only intervention) |  |  | ☑ |  |  |
| Weekly report and questionnaire (only intervention) |  |  | ☑ |  |  |
| Final questionnaire (only intervention) |  |  |  | ☑ |  |
| Semistructured interview (20 participants in intervention group) |  |  |  |  | ☑ |

BMI, body mass index.

adherence score will be calculated averaging adherence rates across all intervention components.

*Evaluation of intervention components*
Using the data from the final online questionnaire, we will calculate means and SDs of the ratings for each intervention component.

### Retention
The daily questionnaire and weekly report emails will act as prompts for the participants to engage with the intervention. At the end of the follow-up visit, participants will receive a £35 one4all gift card.

There are no criteria for withdrawal other than participants' request to withdraw. Participants can also ask to withdraw their collected data. We will ask participants wishing to withdraw whether they are willing to attend the final follow-up and take part in an exit interview to understand the reasons for their discontinuation.

### Statistical analyses
The statistical analysis of the primary outcome, effectiveness of the intervention for weight loss, will be carried out on the basis of intention-to-treat (ITT). That is, after randomisation, participants will be analysed according to their allocated intervention group. We will endeavour to obtain full follow-up data on every participant to allow full ITT analysis. A linear regression, predicting weight at 8 weeks' follow-up while adjusting for baseline weight and GP practice, will assess the effect of condition.[37] We will assess the sensitivity of the analysis to different assumptions about missing data using a variety

of imputation methods, including baseline observation carried forward analysis and an analysis of participants completing follow-up. A final analysis will impute the last home-measured weight for people who did not attend the final follow-up. All tests will be done at a 5% two-sided significance level.

Secondary outcomes will be analysed in several ways. Adherence rates will be assessed and compared between the experimental and control condition. Moderator analyses will assess the effect of adherence, highest educational qualification, liking of weighing at baseline, and overall intervention rating at follow-up (only intervention condition) on weight change.

Further exploratory analyses may be added post-hoc based on preliminary findings. The statistical analysis plan can be found in online supplementary appendix 2.

### Qualitative study
All interview audio-recordings will be transcribed and entered into the NVivo software package (QSR International) for qualitative data analysis. Framework analysis according to Ritchie and Spencer[38] will assess the participants' experiences and perceptions of the different intervention components. The findings will be put into context with the results from the final questionnaire. Inductive thematic analysis following Braun and Clarke[39] will explore additional themes, including barriers and unmet needs. One researcher with training in qualitative methods will perform coding for all interviews.

## Trial management group

The day-to-day management and operation of the study will be coordinated by KF. A Trial Management Group (TMG), consisting of the authors of this paper, will have oversight of the trial. The TMG will be responsible for the monitoring of all aspects of the trial's conduct and progress and will ensure that the protocol is adhered to and that appropriate action is taken to safeguard participants and the quality of the trial itself. The TMG will meet regularly throughout the course of the trial.

## Adverse events

This is a low risk trial where it is implausible that the intervention will lead to differences in the occurrence of adverse events so we decided that it was inappropriate to burden participants to collect and record these.

## Trial monitoring

This is a short trial with no adverse event monitoring or stopping rules so we deemed that a trial steering committee and a data monitoring committee were unnecessary.

## Data management

Data will be kept in accordance with GCP, the Data Protection Act 2018 and General Data Protection Regulation. Two separate databases will be created, one containing all participant identifiable information, the other capturing all outcome data in an anonymised manner, using a unique participant ID. Weight, height and body composition measurements will be entered into the second database by the researcher. Data from the online questionnaires will be downloaded from Qualtrics and added to the second database. The two databases as well as the anonymised recordings and transcriptions from follow-up interviews will be stored on the secure departmental drive and will only be accessible by members of the TMG. After a lay summary of results has been sent out to participants, the database with participant identifiable data will be destroyed. We will retain the anonymised database for future secondary analyses.

Direct access to study data will be granted to authorised representatives from the sponsor for monitoring and/or audit of the study to ensure compliance with regulations. This access, the reason for it and who has authorised it will be recorded by the TMG. Otherwise, confidentiality will be maintained and no-one outside the TMG will have access to the database.

## Ethics and dissemination

Any substantial changes to the protocol will be submitted as an amendment to the NHS National Research Ethics Committee and the Health Research Authority, as well as the Sponsor. On completion of the trial, KF will submit an End of Study notification and final report to these institutions.

We intend to publish the results of this study in peer-reviewed journals, regardless of the nature of the outcome. Authorship will be determined in accordance with the International Committee of Medical Journal Editors (ICMJE) guidelines. We will also present our findings at national and international conferences, and publicise our publications through the departments' online presence. Participants will be informed of the trial results through an information sheet prepared for a lay audience. We will also inform our patients and public involvement panel members about the findings of the study through their regular newsletter.

**Acknowledgements** We would like to acknowledge and thank the members of the public who have helped design the PREVAIL intervention. We thank members of the department who have contributed in expert brainstorming sessions to the development of the intervention. We would like to thank Carmen Piernas and Rhiannon Edwards for creating the randomisation sequence and the randomisation envelopes. We thank the Clinical Trials Unit of the Nuffield Department of Primary Care Health Sciences for their help with setting up this study and support in conducting the trial.

**Contributors** KF, JHB, SJ and PA contributed to the design of the intervention and this study. KF led the preparation of the trial. All authors commented and worked on this paper. The sponsor has reviewed all participant-facing documents as part of the ethics application.

**Funding** This research is funded by the National Institute for Health Research (NIHR) Collaboration for Leadership in Applied Health Research and Care Oxford (CLAHRC) at Oxford Health NHS Foundation Trust. KF's time on this project is funded by NIHR CLAHRC Oxford at Oxford Health NHS Foundation Trust, Wolfson College, University of Oxford (Oxford-Wolfson Marriott-Primary Care Graduate Scholarship), and NIHR School for Primary Care Research (NIHR SPCR). JHB's, SJ's and PA's time on this project is funded by the NIHR Oxford Biomedical Research Centre (BRC) and Oxford CLAHRC. PA and SJ are NIHR senior investigators.

**Disclaimer** The views expressed are those of the authors and not necessarily those of the NHS, the NIHR, the Wellcome Trust or the Department of Health and Social Care. The sponsor is not involved in the collection, management, analysis and interpretation of data; writing of the report; and the decision to submit the report for publication of the study.

**Competing interests** None declared.

**Patient consent for publication** Not required.

**Ethics approval** This trial was reviewed and approved by the NHS National Research Ethics Committee and the Health Research Authority (reference number: 18/SC/0482).

**Provenance and peer review** Not commissioned; externally peer reviewed.

**ORCID iDs**
Kerstin Frie http://orcid.org/0000-0002-4717-5874
Jamie Hartmann-Boyce https://orcid.org/0000-0001-9898-3049
Susan A Jebb https://orcid.org/0000-0001-9190-2920
Paul Aveyard http://orcid.org/0000-0002-1802-4217

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
