## [Reviewer comments · BMJ Open]

ARTICLE DETAILS

TITLE (PROVISIONAL)	Testing the effectiveness of a weight loss intervention to enhance self-regulation in adults who are obese: protocol for a randomised controlled trial
AUTHORS	Frie, Kerstin; Hartmann-Boyce, Jamie; Jebb, Susan A; Aveyard, Paul

VERSION 1 – REVIEW

REVIEWER	Perry Foley Senior Research Associate, OCHIN, Inc., United States of America
REVIEW RETURNED	01-Jun-2019

GENERAL COMMENTS	This is a well-written description of an 8-week randomized controlled trial of a weight-loss intervention aimed to promote self-regulation. A few additional details would enhance the protocol description: - Include dates of study (planned if recruitment has not yet started)- Clarify whether the intervention includes any educational component- On what criteria will the 20 participants asked to complete qualitative interviews be purposively sampled?- Will the control group be somehow discouraged from recording their daily weights? Thank you for the opportunity to review this manuscript and I look forward to reading the results!
--

REVIEWER	Yoshio Nakata University of Tsukuba, Japan
REVIEW RETURNED	18-Oct-2019

GENERAL COMMENTS	The present manuscript is a protocol for a randomised controlled trial to test the effectiveness of a weight loss intervention using self-regulation technique. The manuscript seems well-written and interesting. Several points should be addressed as follows. 1. In abstract (line 65), PREVAIL appeared firstly. Spelling-out or rewording is necessary.2. The rationale of the short intervention period (8 weeks) should be addressed.3. Why do you measure body composition? The necessity should be shown. Validation for the body composition scale (SC-240 MA) should also be shown.4. "111decided" on page 8 (line 42) might be a typo.
---

	5. Is the body scale provided to the intervention group the same as provided to the control group? The scale for the control group is clear (BodyTrace, Inc., New York) but not for the intervention group. The scale can automatically transfer measurements to a server and the researchers can access the database. Is this function also used for the intervention group? 6. In the interview after the intervention, how do you select 20 participants? 7. In Ethics and dissemination (page 16), REC, HRA, and PPI might be spelled out.
--	---

VERSION 1 – AUTHOR RESPONSE

Reviewer: 1

Reviewer Name: Perry Foley

Institution and Country: Senior Research Associate, OCHIN, Inc., United States of America

Please state any competing interests or state 'None declared': None declared

This is a well-written description of an 8-week randomized controlled trial of a weight-loss intervention aimed to promote self-regulation. A few additional details would enhance the protocol description:

- Include dates of study (planned if recruitment has not yet started)

Response: We agree that the dates of study would be useful to add. The study ran between April and October 2019. We have added this information in the methods section.

- Clarify whether the intervention includes any educational component

Response: We do not educate participants on how to lose weight face-to-face. Arguably, however, the manual we provide at the baseline meeting constitutes an educational component, as it explains the rationale of self-regulation and experimentation to intervention group participants, and gives tips on how to best weigh on a daily basis. The manual also provides participants with some background information on the different weight loss actions of the action planning task. We have made this clearer in the methods section of the paper.

- On what criteria will the 20 participants asked to complete qualitative interviews be purposively sampled?

Response: The criteria used to sample the 20 participants are levels of adherence to the intervention, weight change, and responses to the final questionnaire. The lead researcher will choose participants with different levels on these criteria in order to create a more representative sample. We have amended the protocol to be clearer on this point.

- Will the control group be somehow discouraged from recording their daily weights?

Response: In the control group, we ask participants to simply weigh on a daily basis and see what the weighing motivates them to do. We do not give instructions to record daily weight measurements, but we also do not actively discourage participants from doing so. This is because we want to sustain the belief that daily weighing on its own might be an intervention.

In a previous study, less than half of the participants decided to track their weight measurements without prompting (Frie, Hartmann-Boyce, Pilbeam, Jebb, & Aveyard, 2019). When they did track their weight measurements, it was usually on a piece of paper, rather than using an advanced and visualising tool such as the weight tracking app of the intervention condition. We are therefore confident that the control and intervention conditions will differ regarding weight-tracking behaviours.

We have added a sentence in the methods section to clarify that the control group will not be given any instructions other than self-weighing.

Reviewer: 2

Reviewer Name: Yoshio Nakata

Institution and Country: University of Tsukuba, Japan

Please state any competing interests or state 'None declared': None declared.

The present manuscript is a protocol for a randomised controlled trial to test the effectiveness of a weight loss intervention using self-regulation technique. The manuscript seems well-written and interesting. Several points should be addressed as follows.

1. In abstract (line 65), PREVAIL appeared firstly. Spelling-out or rewording is necessary.

Response: We thank the reviewer for this comment. We have amended the abstract to say "self-regulation intervention".

2. The rationale of the short intervention period (8 weeks) should be addressed.

RESPONSE: We thank the reviewer for this comment. Other weight loss studies (e.g. Lally, Chipperfield, & Wardle, 2008) have also used a time frame of eight weeks, we are therefore confident that this study duration is sufficient to assess early effectiveness. If the intervention is acceptable, feasible and shows early effectiveness, a longer trial will be needed to assess long-term effectiveness for weight control. However, until we have such evidence we do not consider a longer trial to be a good use of resources, nor helpful for participants who may be prevented from accessing a more effective intervention by their participation in the study. We have clarified this point in the methods section of the paper.

3. Why do you measure body composition? The necessity should be shown. Validation for the body composition scale (SC-240 MA) should also be shown.

Response: The scales we decided to use for the baseline and follow-up meetings measured body composition automatically. We are not going to use the body composition measurements in the analysis so we have decided to now delete this aspect from the protocol to avoid confusion.

4. "111decided" on page 8 (line 42) might be a typo.

Response: We thank the reviewer for spotting this typo, we have removed it from the revised version of the protocol.

5. Is the body scale provided to the intervention group the same as provided to the control group? The scale for the control group is clear (BodyTrace, Inc., New York) but not for the intervention group. The scale can automatically transfer measurements to a server and the researchers can access the database. Is this function also used for the intervention group?

Response: The intervention group participants receive different and simpler digital weighing scales, from the company Etekcity Corporation (California, USA). The BodyTrace scales we use for the control condition are costly, and our budget did not allow for us to buy these scales for all 100 participants. As participants in the control condition do not track their weight measurements, the BodyTrace scales are necessary to assess self-weighing adherence in in this group. In the intervention group, since we ask participants to track their weight measurements, we can assess

adherence to daily weighing through other measures, and therefore do not need BodyTrace scales. We recognise that differences between self-reported and non-self-reported weight measurements have repeatedly been found, and we will address this as a limitation in our results paper. We have amended the protocol to be clearer on the use of different scales in the two conditions.

6. In the interview after the intervention, how do you select 20 participants?

Response: The lead researcher will choose participants with different levels of adherence, weight change and liking of intervention features, in order to create a more representative sample. We have amended the protocol to be clearer on this point.

7. In Ethics and dissemination (page 16), REC, HRA, and PPI might be spelled out.

Response: We thank the reviewer for the comment. We have added the abbreviations in brackets where REC, HRA and PPI are first spelled out in the protocol.

References

Frie, K., Hartmann-Boyce, J., Pilbeam, C., Jebb, S., & Aveyard, P. (2019). Analysing self-regulatory behaviours in response to daily weighing: a think-aloud study with follow-up interviews. *Psychol Health*, 1-20. doi:10.1080/08870446.2019.1626394

Lally, P., Chipperfield, A., & Wardle, J. (2008). Healthy habits: efficacy of simple advice on weight control based on a habit-formation model. *International journal of obesity*, 32(4), 700-707. doi:10.1038/sj.ijo.0803771

VERSION 2 – REVIEW

REVIEWER	Yoshio Nakata University of Tsukuba, Japan
REVIEW RETURNED	10-Nov-2019
GENERAL COMMENTS	I confirm that the authors have done what was asked. The revised manuscript is acceptable.